

# RANK signaling in osteoclast precursors results in a more permissive epigenetic landscape and sexually divergent patterns of gene expression

Abigail L. Keever[1,2,*], Kathryn M. Collins[2,*], Rachel A. Clark[2], Amber L. Framstad[2] and Jason W. Ashley[2]

[1] Elson S. Floyd College of Medicine, Washington State University, Spokane, WA, United States
[2] Department of Biology, Eastern Washington University, Cheney, WA, United States
* These authors contributed equally to this work.

Corresponding author
Jason W. Ashley, jashley6@ewu.edu

## ABSTRACT

**Background:** Sex is an important risk factor in the development of osteoporosis and other bone loss disorders, with women often demonstrating greater susceptibility than men. While variation in sex steroids, such as estradiol, accounts for much of the risk, there are likely additional non-endocrine factors at transcriptional and epigenetic levels that result in a higher rate of bone loss in women. Identification of these factors could improve risk assessment and therapies to preserve and improve bone health.

**Methods:** Osteoclast precursors were isolated male and female C57Bl/6 mice and cultured with either MCSF alone or MCSF and RANKL. Following the culture period RNA was isolated for RNA sequencing and DNA was isolated for tagmentation and ATAC sequencing. RNA-Seq and ATAC-seq were evaluated *via* pathway analysis to identify sex- and RANKL-differential transcription and chromatin accessibility.

**Results:** Osteoclasts demonstrated significant alterations in gene expression compared to macrophages with both shared and differential pathways between the sexes. Transcriptional pathways differentially regulated between male and female cells were associated with immunological functions with evidence of greater sensitivity in male macrophages and female osteoclasts. ATAC-Seq revealed a large increase in chromatin accessibility following RANKL treatment with few alterations attributable to sex. Comparison of RNA-Seq and ATAC-seq data revealed few common pathways suggesting that many of the transcriptional changes of osteoclastogenesis occur independently of chromatin remodeling.

## INTRODUCTION

Osteoclasts are multinucleated hematopoietic-lineage cells responsible for the dissolution of bone matrix required for both physiological bone growth and remodeling (*Teitelbaum, 2007*). While indispensable for normal bone homeostasis, excessive osteoclastic resorption can lead to insufficient bone mineral density, joint destruction, tooth loss, and fracture (*Ensrud & Crandall, 2017*). Multiple genetic and environmental factors contribute to

overall risk of pathologically elevated osteoclast activity, but the greatest contributors are advanced age and female sex (*Watts et al., 2008*; *Johannes de Villiers, 2009*). Osteoporosis, a generalized decrease in bone mass to at least 2.5 standard deviations below that of an average young adult, results in a lifetime risk of fracture of one in five among men over fifty and one in three among women over fifty (*De castro machado, Hannon & Eastell, 1999*; *Yelin, Weinstein & King, 2016*; *Sebbag et al., 2019*). The high rate of osteoporosis resulting in both direct medical costs and loss of productivity and quality of life is an impetus for improving understanding of additional risk factors and identifying new therapeutic targets.

The disproportionate osteoporosis risk among women is explained mostly by the regulatory role of estrogens in osteoclast differentiation and function. Estradiol limits resorption by both reducing differentiation and limiting the lifespan of osteoclasts (*Shevde et al., 2000*; *Krum et al., 2008*). Prior to menopause, ovary-derived estradiol protects young female bone mass. After menopause, which is marked by cessation of ovarian follicle development and a precipitous decline in circulating estradiol, osteoclasts can become disinhibited. If the hyperactivity of postmenopausal osteoclasts cannot be overcome by increased bone-forming activity of osteoblasts, progressive bone loss and osteoporosis follows (*Recker et al., 2004*; *Iki et al., 2004*).

Men experience osteoporosis risk differently. Although circulating estradiol level in men of all ages is comparable to that of post-menopausal women, aromatase enzyme (encoded by the CYP19A1 gene) present in the bone converts testosterone to estradiol locally, where it exerts its protective effect (*Sasano et al., 1997*; *Gennari, Nuti & Bilezikian, 2004*; *Khosla et al., 2005a*, *2005b*). In contrast to women and estradiol, men do not experience a rapid decrease in testosterone at an age-dependent milestone like menopause, but rather demonstrate increasing variation beginning in the second decade of life (*Kelsey et al., 2014*). Sex-specific endocrine differences likely account for much of sex-dependent osteoporosis risk, with bone parameters of men in their eighth decade approaching parity with women in their sixth (*Alswat, 2017*).

While estrogen deficiency explains much of the sex-differential osteoporosis risk, the regulators of osteoclast differentiation and activity are diverse, and there are likely sexually divergent contributors independent of sex steroids. Osteoclasts differentiate from macrophage-like precursors *via* stimulation with Receptor Activator of Nuclear Factor κB Ligand (RANKL). Upon binding of RANKL to its cognate receptor, RANK, Tumor necrosis factor Receptor Associated Factor (TRAF)-dependent signaling cascades converge on multiple transcription factors including JNK, ERK, p38, and NFκB (*Feng, 2005*). This receptor-proximal signaling alters expression of multiple genes; chief among them is upregulation of Nuclear Factor of Activated T Cells 1 (NFATc1), which drives much of the osteoclast differentiation program (*Kim & Kim, 2014*, p. 1). Expression of NFATc1 and other osteoclastic genes is also subject to epigenetic regulation (*Rohatgi et al., 2018*; *Das et al., 2018*; *Shin et al., 2019*; *Astleford et al., 2020*). The complexity of osteoclastic gene expression presents opportunities for variation in degree of differentiation and level of resultant resorptive activity, and multiple pathways other than RANKL/RANK can modulate osteoclastogenesis (*Liu et al., 2009*; *Goel et al., 2019*). From this we hypothesized

that there are sexually divergent patterns of gene regulation in the differentiation of osteoclasts.

Here we report the findings of our analyses of gene expression and chromatin accessibility in bone marrow macrophages and osteoclasts extracted from female and male mice.

## MATERIALS AND METHODS

### Osteoclast precursor culture

Mice used as cell sources were euthanized and provided by the staff of the Eastern Washington University vivarium (PHS animal welfare assurance number D19-01059). Primary mouse bone marrow cells were obtained by flushing α-MEM medium (Minimum Essential Medium Eagle (M0894; MilliporeSigma, St. Louis, MO, USA), 20 mM GlutaMAX (35050061; Thermo Fisher, Waltham, MA, USA), and 10% heat-inactivated fetal bovine serum (F-500-D; Atlas Biologicals, Fort Collins, CO, USA)) through the femurs and tibias of 3-month-old C57Bl6 female and male mice (three each for RNA-sequencing; two each for ATAC-Sequencing) (Kaur et al., 2017). Bone marrow cells were cultured at 37 °C and 5% $CO_2$ overnight in tissue culture-treated dishes to allow adherent cells to attach. The next day, non-adherent cells were transferred to non-treated suspension culture dishes and supplemented with 25 ng/mL Macrophage Colony-Stimulating Factor/MCSF (200-08; Shenandoah Biotechnology, Warminster, PA, USA) to induce the differentiation of macrophages, which can attach even to non-treated culture surfaces. After 24 h, the MCSF-containing medium was refreshed to remove non-macrophage cells, and macrophages were maintained for an additional 48 h to allow them to proliferate prior to experimental treatments.

Osteoclasts were TRAP stained according to the University of Rochester Center for Musculoskeletal Research protocol ("Forms and Protocols—Histology Core—Core Services—Center for Musculoskeletal Research—University of Rochester Medical Center", https://www.urmc.rochester.edu/musculoskeletal-research/core-services/histology/protocols.aspx). Cells were fixed in a solution of 8% neutral buffered formalin (10%, 10790-714; VWR, Radnor, PA, USA), 26% citrate solution (915-50ML, Millipore-Sigma, St. Louis, MO, USA), and 66% acetone (534064-500ML, Millipore-Sigma, St. Louis, MO, USA) for 2 min and washed twice with phosphate buffered saline before a 1 h incubation at 37 °C in a pH 5.0 aqueous solution of 9.2 g/L sodium acetate (S-2889; MilliporeSigma, St. Louis, MO, USA), 11.4g/L L-(+) tartaric acid (T-6521, MilliporeSigma, St. Louis, MO, USA), 2.8 mL/L glacial acetic acid (AX0074-6; MilliporeSigma, St. Louis, MO, USA), 100 mg/L Napthol AS-MX Phosphate (N-4875; MilliporeSigma, St. Louis, MO, USA; freshly dissolved in 2-ethoxyethanol (E-2632; MilliporeSigma, St. Louis, MO, USA) at a concentration of 20 mg/mL), and 600 mg/L Fast Red Violet LB Salt (F-3381; MilliporeSigma, St. Louis, MO, USA). After staining, wells were washed twice with deionized water and allowed to air dry.

## RNA-Sequencing (RNA-Seq)

Appropriate sample sizes were calculated using RNASeqPower with the following parameters: average read depth = 20 million, biological coefficient of variation = 0.1, effect size = 2, alpha = 0.05, power = 0.9 (*Hart et al., 2013*; *Therneau & Stephen, 2022*). From this, a sample size of three measurements per group were determined as sufficient. Osteoclast precursors from male and female mice were divided into two treatment groups: one group of cells (macrophages) were maintained in α -MEM with 25 ng/mL MCSF for 72 h; the other group of cells (osteoclasts), were maintained in α -MEM with 25 ng/mL MCSF (200-08; Shenandoah Biotechnology, Warminster, PA, USA) and 100 ng/mL RANKL (200-04; Shenandoah Biotechnology, Warminster, PA, USA) for 72 h. Media were refreshed at 48 h. At the conclusion of the culture period, cells were lysed in TRI reagent and RNA was extracted using the Direct-zol RNA miniprep kit (R2051; Zymo Research, Irvine, CA, USA). RNA was extracted from each group (female-macrophages, female-osteoclasts, male-macrophages, and male-osteoclasts) in triplicate. RNA samples were transferred to GeneWiz/Azenta for cDNA synthesis, indexing, sequencing, and basic RNA-seq analysis (de-multiplexing, alignment, transcript identification, and differential expression analysis). Raw sequence reads were trimmed to remove adaptors and poor sequence quality nucleotides using Trimmomatic v.0.36, trimmed reads were mapped to the GRCm38 *Mus musculus* reference genome using STAR aligner v.2.5.2b, unique gene hit counts from reads falling within exon regions were calculated with the featureCounts module of Subread package v.1.5.2, and differential expressions were calculated using DESeq2 with *p*-values and log2 fold changes determined *via* the Wald test. Differential expression data is included as Supplemental Files. RNA sequencing data is available at NCBI GEO, accession number GSE216929

## Assay for transposase-accessible chromatin-sequencing

Osteoclast precursors from male and female mice were divided into two groups: one group of cells (MCSF only) were maintained in α-MEM with 25 ng/mL MCSF for 24 h; the other group of cells (MCSF+RANKL),were maintained in α-MEM with 25 ng/mL MCSF and 100 ng/mL RANKL for 24 h. At the conclusion of the culture period, nuclei were isolated and used in tagmentation and indexing reactions using the ATAC-Seq Kit (53150, Active Motif, Carlsbad, CA, USA). ATAC-seq libraries were prepared from each sex and treatment group (female-MCSF, female-MCSF+RANKL, male-MCSF, male-MCSF +RANKL) in duplicate. ATAC-seq libraries were transferred to GeneWiz/Azenta, which performed the sequencing reactions and basic ATAC-seq analysis (de-multiplexing, alignment, peak calling, and peak differential analysis). Sequencing adaptors and low-quality nucleotides were removed with Trimmomatic v.0.38, and reads were aligned to the *Mus musculus* mm10 reference genome using bowtie2. Aligned reads were filtered with samtools v1.9 to preserve alignments with a minimum mapping quality of 30, are aligned concordantly, and are the primary called alignments; PCR or digital duplicates were marked with Picard v2.18.26 and removed. Mitochondrial reads and reads mapped to unplaced contigs were also removed. Open chromatin regions were identified through peak calling with MACS2 v2.1.2, and consensus peaks from sample duplicates were

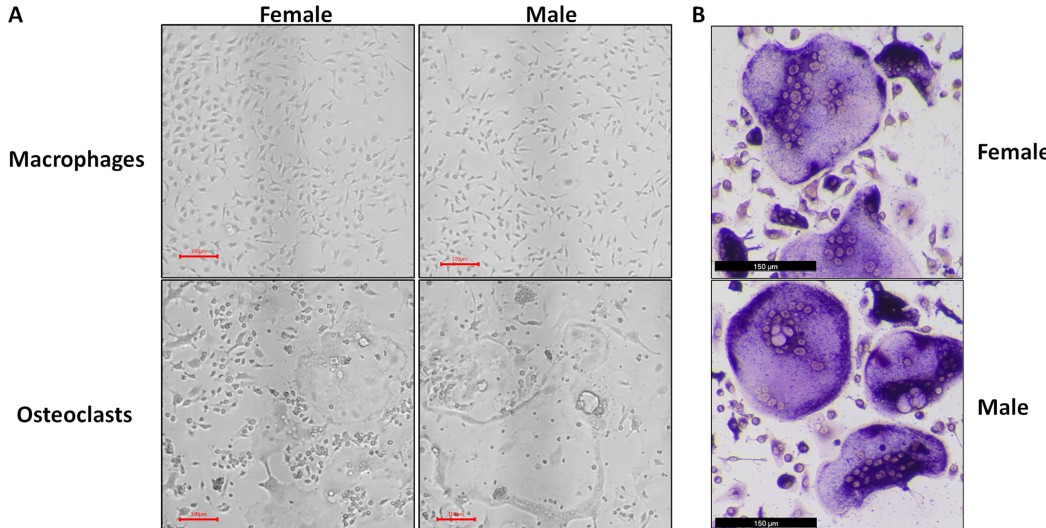

**Figure 1 Representative micrographs of undifferentiated macrophages and osteoclasts.** Male and female bone marrow-derived macrophages were cultured with 25 ng/mL MCSF alone or 25 ng/mL MCSF and 100 ng/mL RANKL for 3 days. (A) Micrographs of macrophages and osteoclasts prior to lysis and RNA extraction. Cells cultured with RANKL demonstrated multinucleation and increased size. (B) Micrographs of TRAP-stained osteoclast differentiated under conditions matched to RNA source cells.

merged and kept for downstream analysis. After reads falling beneath peaks were counted, differential peak calling (corresponding to differentially accessible chromatin regions) was performed using the R package Diffbind. Differential peak call data are provided as Supplemental Files.

## Pathway analyses of differential sequencing data

Differential data with non-adjusted *p*-values were uploaded to iPathwayGuide (Advaita Bioinformatics; http://www.advaitabio.com/ipathwayguide). iPathwayGuide is a web-based bioinformatics platform that applies Impact Analysis to differential datasets, which considers both over-representation of differential genes in a particular pathway and computed perturbation to generate pathway specific *p*-values (*Tarca et al., 2009*; *Ahsan & Drăghici, 2017*). *p*-Values for individual genes were adjusted *via* False Discovery Rate. Venn diagrams were generated with eulerr.co (*Larsson, Godfrey & Gustafsson, 2021*).

# RESULTS

## Gene expression patterns in macrophages and osteoclasts cluster by sex and differentiation state

At the conclusion of the treatment period, cells cultured with MCSF alone retained the typical fusiform morphology of immunologically naïve macrophages, where MCSF and RANKL treatment resulted in large, multinucleated osteoclasts (Fig. 1A). Osteoclasts differentiated under matched conditions were TRAP positive (Fig. 1B). Following RNA-seq, principal component analysis of replicates from each group revealed that in both sexes, the principal component of clustering (PC1) was differentiation state (Fig. 2A).
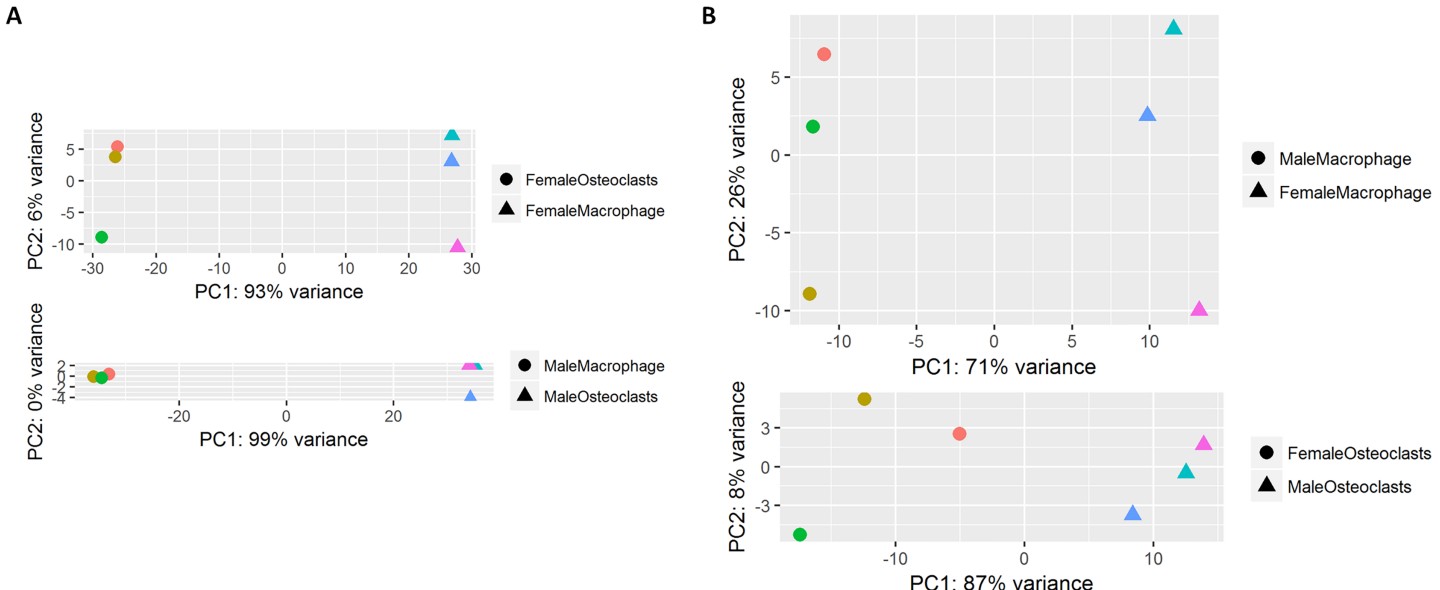

**Figure 2 Principal component analysis of RNA-Seq data.** (A) RNA-seq replicates from male and female samples demonstrate PC1 clustering according to differentiation state. (B) RNA-seq replicates from macrophages and osteoclasts demonstrate PC1 clustering according to sex.

Similarly, when comparing cells of the same differentiation state, sex of the cells was the principal component (Fig. 2B).

## RNA-sequencing of macrophages and osteoclasts identifies expected differential gene expression

Figure 3 depicts biclustering analyses of 30 differentially expressed genes with the lowest adjusted *p*-values according to differentiation state and sex. Among both male and female cells, matrix metalloproteinase 9 (Mmp9), cathepsin K (Ctsk), and tartrate-resistant acid phosphatase (Acp5), which are involved in osteoclast function, were among the most significantly differential (Fig. 3A). When comparing cells of the same differentiation state and opposite sexes, Ubiquitously Transcribed Tetratricopeptide Repeat Containing, Y-Linked (Uty), DEAD-Box Helicase 3 Y-Linked (Ddx3y), and Eukaryotic translation initiation factor 2 subunit 3, Y-linked (Eif2s3y), which are restricted to the Y chromosome, were found to be more highly expressed in male cells. Conversely, X-inactive specific transcript (Xist), which is expressed only in cells with more than one X chromosome, was more highly expressed in female cells (Fig. 3B). The identification of expected differences in gene expression by the unbiased RNA-seq method increases confidence in the method and subsequent analyses to identify novel differential gene expression based on sex and differentiation state.

## Sex-independent and sex-dependent differential pathways in osteoclastogenesis

Using a *p*-value threshold of 0.001 and a minimum fold change of 1.5, 2,280 differentially expressed genes were identified in female cells and 2,844 in male (Fig. 4A). Following false

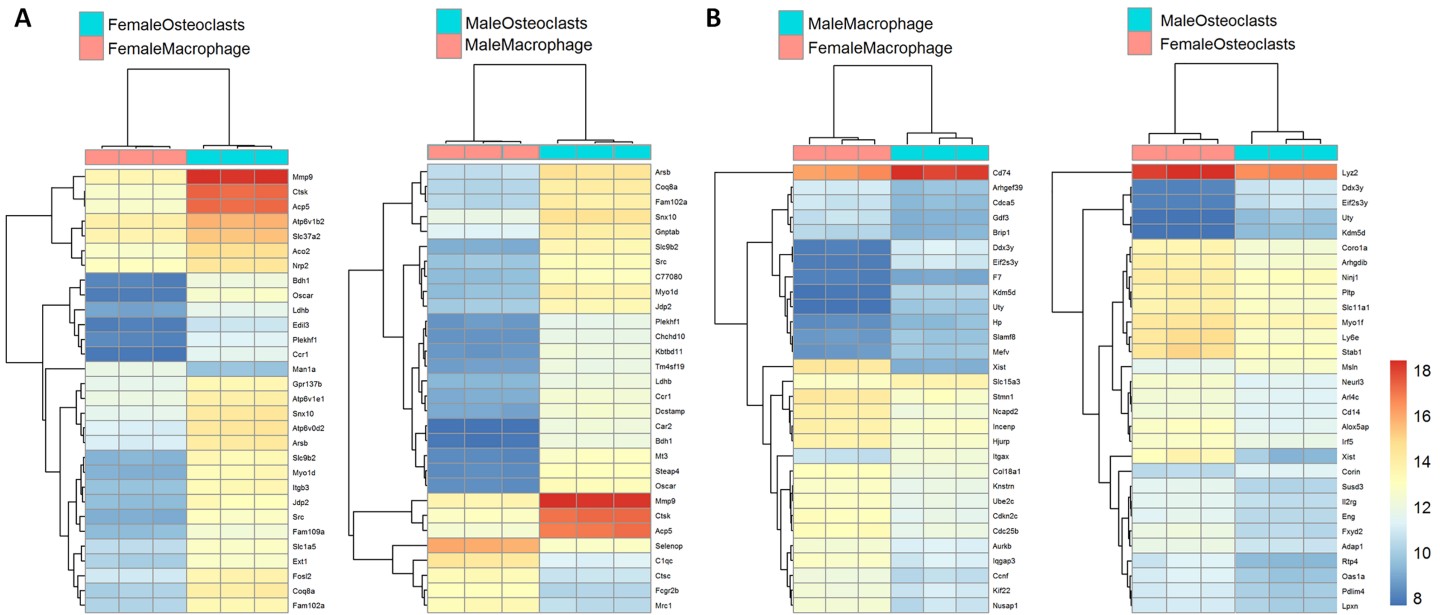

**Figure 3 Biclustering analyses of top 30 differentially expressed genes.** (A) Differentially regulated genes in male and female cells by differentiation state. In both male and female cells, osteoclastic genes MMP9, Ctsk, and Acp5 demonstrate the most significant differences in expression. (B) Differentially regulated genes in macrophages and osteoclasts by sex. Expected female (XIST) and male (Ddx3y, Uty, and Eif2s3y) differential genes were identified in both cell types.

discovery rate correction, pathway analysis of differential gene expression according to differentiation state identified 22 altered pathways common to both female and male cells, 45 specific to male cells, and two specific to female cells (Fig. 4B). These pathways are listed in Table 1.

## Differentiation-independent and differentiation-dependent differential pathways in female and male cells

Using a $p$-value threshold of 0.05 and a minimum fold change of 0.6, 1,347 differentially expressed genes were identified in macrophages and 1,618 in osteoclasts (Fig. 5A). Following false discovery rate correction, pathway analysis of differential gene expression according to sex identified 32 altered pathways common to both macrophages and osteoclasts, 15 specific to macrophages, and 34 specific to osteoclasts (Fig. 5B). These pathways are listed in Table 2.

## Differential transposase accessible chromatin regions regulated by RANKL in female and male cells

ATAC-seq revealed that most differences in transposase-accessible chromatin were in promotor and distal intergenic regions (Fig. 6). Using a $p$-value threshold of 0.01 and a minimum accessibility fold difference of 2.7 (female) or 2.4 (male), there were 4,957 genes with differentially accessible chromatin in female MCSF+RANKL-treated cells and 4,994 in male (Fig. 7A). Most differentially accessible genes were more accessible following MCSF+RANKL treatment with more accessible genes accounting for 95% in female cells and 94% in male cells. Without FDR correction, there were 17 differentially accessible

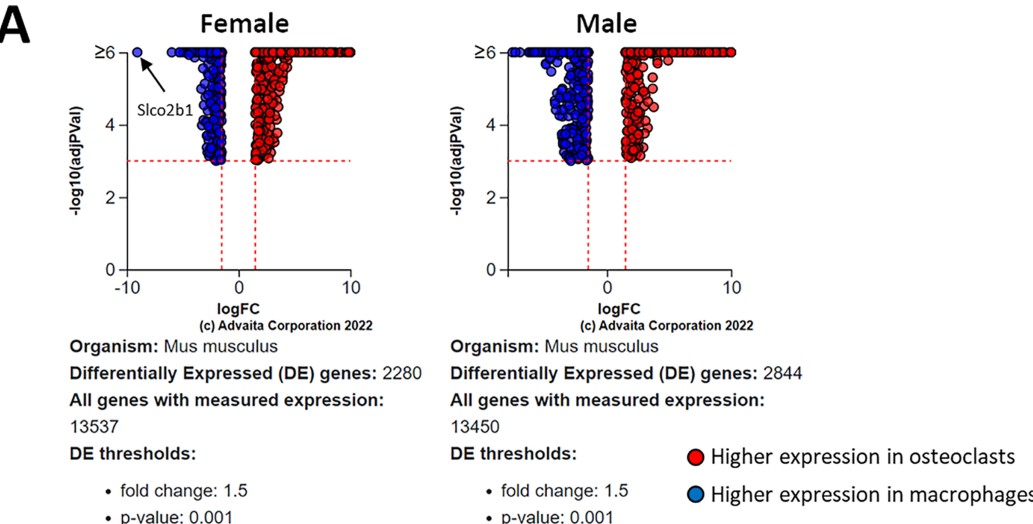

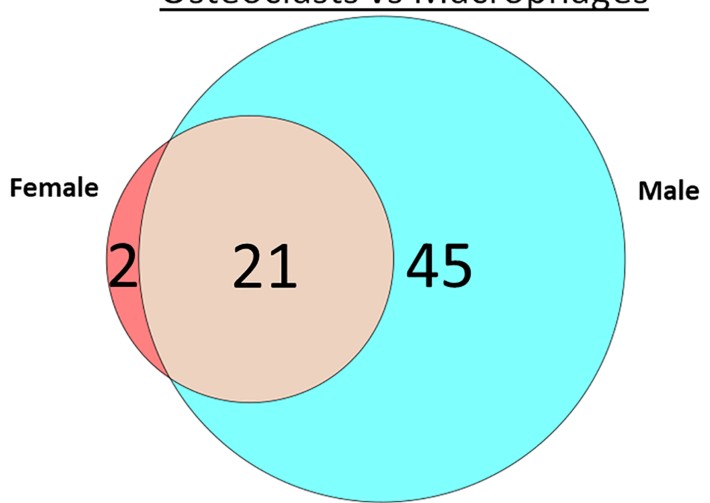

**Figure 4 Gene and pathway analysis of osteoclastogenesis RNA-Seq data.** (A) Volcano plots depicting genes meeting fold change and *p*-value thresholds in female and male cells. Differentials with −log10 transformed adjusted *p*-values of six and greater are plotted at the same level. Full volcano plots are included in Fig. S1. (B) Female-specific, male-specific, and common pathways altered during osteoclastogenesis. Pathways are listed in Table 1.

pathways specific to female cells, six specific to male cells, and five common to both sexes (Fig. 7B). These pathways are listed in Table 3.

### Sex-independent and sex-dependent differential pathways in differential transposase accessible chromatin regions in MCSF- and MCSF+RANKL-treated cells

Using a *p*-value threshold of 0.05 and a minimum accessibility fold difference of 0.6, there were 84 genes with differentially accessible chromatin in MCSF-treated cells and 122 in

**Table 1 Altered transcriptional pathways during osteoclastogenesis.**

| Female-specific | | Male-specific | | Sex-independent | | |
|---|---|---|---|---|---|---|
| Pathway | *p*-value | Pathway | *p*-value | Pathway | *p*-value (Female) | *p*-value (Male) |
| Amyotrophic lateral sclerosis | 9.91E−05 | Cytokine-cytokine receptor interaction | 1.82E−06 | Non-alcoholic fatty liver disease | 8.23E−05 | 5.48E−05 |
| Taste transduction | 0.016 | Viral protein interaction with cytokine and cytokine receptor | 1.82E−06 | Huntington disease | 9.91E−05 | 0.009 |
| | | Antigen processing and presentation | 1.28E−05 | Thermogenesis | 1.98E−04 | 3.63E−04 |
| | | *Staphylococcus aureus* infection | 1.73E−05 | Retrograde endocannabinoid signaling | 1.98E−04 | 0.007 |
| | | Influenza A | 2.03E−05 | Cardiac muscle contraction | 2.06E−04 | 9.33E−05 |
| | | Allograft rejection | 5.09E−05 | Citrate cycle (TCA cycle) | 2.06E−04 | 9.33E−05 |
| | | Graft-*versus*-host disease | 5.09E−05 | Collecting duct acid secretion | 2.06E−04 | 9.33E−05 |
| | | Intestinal immune network for IgA production | 5.09E−05 | Metabolic pathways | 2.06E−04 | 9.33E−05 |
| | | Toll-like receptor signaling pathway | 5.09E−05 | Oxidative phosphorylation | 2.06E−04 | 9.33E−05 |
| | | Type I diabetes mellitus | 5.09E−05 | Parkinson disease | 2.06E−04 | 0.006 |
| | | Autoimmune thyroid disease | 6.40E−05 | Pathways of neurodegeneration-multiple diseases | 2.06E−04 | 0.046 |
| | | Inflammatory bowel disease | 7.05E−05 | Prion disease | 2.68E−04 | 6.78E−05 |
| | | Chemokine signaling pathway | 8.80E−05 | Alzheimer disease | 2.68E−04 | 0.009 |
| | | Hematopoietic cell lineage | 9.33E−05 | Synaptic vesicle cycle | 7.17E−04 | 3.63E−04 |
| | | Viral myocarditis | 9.33E−05 | Phagosome | 0.001 | 9.33E−05 |
| | | Malaria | 9.46E−05 | Neuroactive ligand-receptor interaction | 0.002 | 5.09E−05 |
| | | Tuberculosis | 1.38E−04 | Rheumatoid arthritis | 0.002 | 8.80E−05 |
| | | Asthma | 1.52E−04 | Calcium signaling pathway | 0.002 | 3.09E−04 |
| | | Measles | 3.63E−04 | Axon guidance | 0.003 | 0.004 |
| | | NOD-like receptor signaling pathway | 4.40E−04 | Cell adhesion molecules | 0.005 | 9.33E−05 |
| | | Bile secretion | 0.002 | Systemic lupus erythematosus | 0.022 | 1.87E−05 |
| | | Pertussis | 0.002 | | | |
| | | Osteoclast differentiation | 0.003 | | | |
| | | Hepatitis B | 0.004 | | | |
| | | Human cytomegalovirus infection | 0.005 | | | |
| | | Legionellosis | 0.007 | | | |
| | | Pathways in cancer | 0.007 | | | |
| | | Arachidonic acid metabolism | 0.009 | | | |
| | | cAMP signaling pathway | 0.009 | | | |
| | | Toxoplasmosis | 0.009 | | | |
| | | Lysosome | 0.011 | | | |
| | | Th17 cell differentiation | 0.011 | | | |
| | | Th1 and Th2 cell differentiation | 0.012 | | | |
| | | MAPK signaling pathway | 0.013 | | | |

(Continued)

| Female-specific | | Male-specific | | Sex-independent | | |
|---|---|---|---|---|---|---|
| Pathway | *p*-value | Pathway | *p*-value | Pathway | *p*-value (Female) | *p*-value (Male) |
| | | C-type lectin receptor signaling pathway | 0.014 | | | |
| | | Pyruvate metabolism | 0.018 | | | |
| | | Necroptosis | 0.02 | | | |
| | | Human papillomavirus infection | 0.021 | | | |
| | | ABC transporters | 0.024 | | | |
| | | Epstein-Barr virus infection | 0.035 | | | |
| | | Herpes simplex virus 1 infection | 0.042 | | | |
| | | Rap1 signaling pathway | 0.043 | | | |
| | | cGMP-PKG signaling pathway | 0.044 | | | |
| | | NF-kappa B signaling pathway | 0.045 | | | |
| | | TNF signaling pathway | 0.046 | | | |

MCSF+RANKL-treated cells (Fig. 8A). Without FDR correction, there were 30 differentially accessible pathways in MCSF-treated cells (Fig. 8B). These pathways are listed in Table 4.

## Differential gene expression and gene accessibility during osteoclastogenesis

Meta-analysis of macrophage *vs* osteoclast RNA-seq and MCSF *vs* MCSF+RANKL ATAC-seq data revealed 51 gene expression specific pathways, 17 gene accessibility specific pathways, and five pathways common to both in female cells (Fig. 9A). These pathways are listed in Table 5. Among male cells, there were 89 gene expression specific pathways, seven gene accessibility specific pathways, and four common pathways (Fig. 9B). These pathways are listed in Table 6.

## DISCUSSION

In this study, we applied high-throughput sequencing and bioinformatic analyses to characterize the transcriptional profile and epigenetic landscape of osteoclastogenesis in cells derived from female and male mice. Throughout this study, we focused our analyses on differences related to sex and differentiation state with a goal of identifying novel genes and pathways that may influence osteoclast differentiation in a sexually divergent manner. Our initial analyses correctly identified well-established genes involved in osteoclast function as well as known sexually divergent genes such as XIST (overexpressed in XX cells) and Uty (found only on Y chromosomes). That our unbiased analyses correctly identified anticipated variation at both the transcriptional and epigenetic level supports their ability to identify novel differences as well.

To manage the large datasets produced in this study, we utilized iPathwayGuide, which applies Impact Analysis to assign genes to pathways described by the Kyoto Encyclopedia

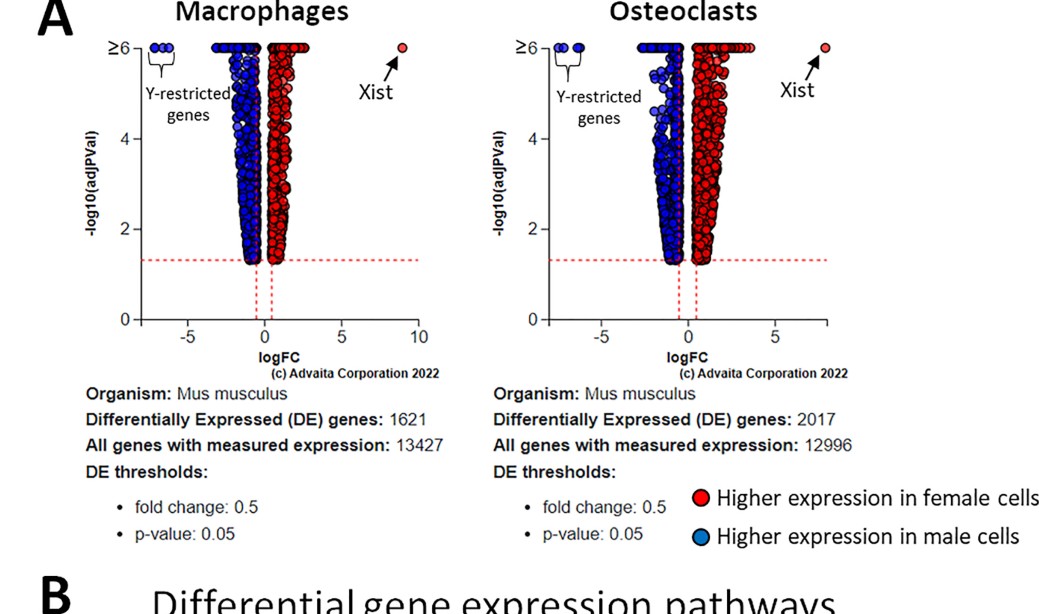

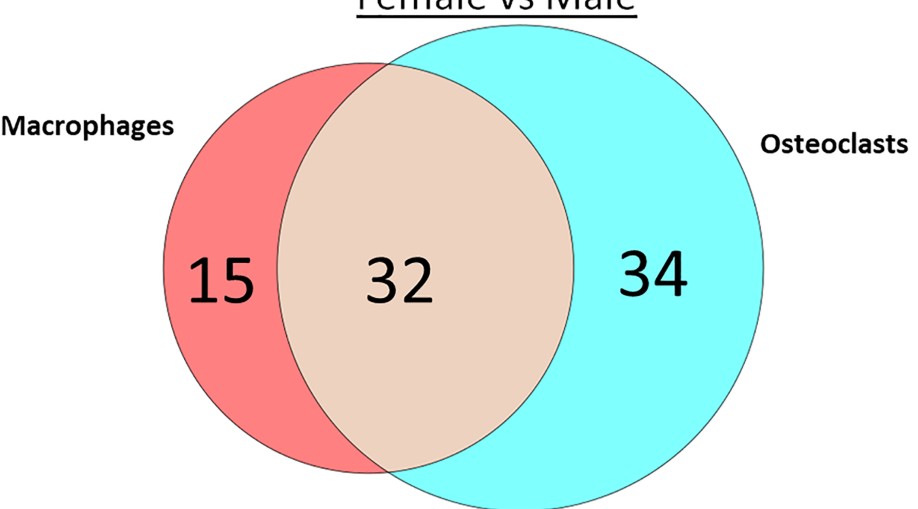

**Figure 5  Gene and pathway analysis of sexually divergent RNA-Seq data.** (A) Volcano plots depicting genes meeting fold change and *p*-value thresholds in macrophages and osteoclasts. Differentials with −log10 transformed adjusted *p*-values of 6 and greater are plotted at the same level. Full volcano plots are included in Fig. S1. (B) Macrophage-specific, osteoclast-specific, and common pathways altered during osteoclastogenesis. Pathways are listed in Table 2.

of Genes and Genomes (KEGG) and statistically evaluates pathways according to fold changes and *p*-values of differentially expressed genes (*Tarca et al., 2009*; *Ahsan & Drăghici, 2017*). We deliberately selected low fold change thresholds for differentially expressed genes when performing pathway analyses for two reasons: first, there is evidence that small fold changes in gene expression can result in significant biological impacts and setting fold change thresholds too high can lead to disposal of potentially relevant genes, and second, by providing more genes to iPathwayGuide, the analysis was better equipped

**Table 2 Altered transcriptional pathways between sexes.**

| Macrophage-specific | | Osteoclast-specific | | Differentiation-independent | | |
|---|---|---|---|---|---|---|
| Pathway | *p*-value | Pathway | *p*-value | Pathway | *p*-value (macrophage) | *p*-value (osteoclast) |
| Fanconi anemia pathway | 2.09E−05 | Measles | 8.40E−05 | Cytokine-cytokine receptor interaction | 9.46E−06 | 5.16E−05 |
| Cell cycle | 7.15E−05 | Inflammatory bowel disease | 2.23E−04 | Viral protein interaction with cytokine and cytokine receptor | 2.09E−05 | 1.24E−04 |
| Asthma | 2.19E−04 | Pertussis | 2.23E−04 | *Staphylococcus aureus* infection | 2.19E−04 | 2.49E−05 |
| DNA replication | 2.19E−04 | Kaposi sarcoma-associated herpesvirus infection | 2.44E−04 | Cell adhesion molecules | 2.19E−04 | 2.02E−04 |
| Homologous recombination | 2.19E−04 | Axon guidance | 6.32E−04 | Systemic lupus erythematosus | 2.19E−04 | 2.44E−04 |
| IL-17 signaling pathway | 0.002 | Human cytomegalovirus infection | 6.32E−04 | Hematopoietic cell lineage | 2.19E−04 | 2.82E−04 |
| Progesterone-mediated oocyte maturation | 0.011 | Pathways in cancer | 0.001 | Leishmaniasis | 2.19E−04 | 0.002 |
| Fluid shear stress and atherosclerosis | 0.015 | MAPK signaling pathway | 0.002 | Rheumatoid arthritis | 2.41E−04 | 5.52E−04 |
| Cellular senescence | 0.021 | ABC transporters | 0.003 | Malaria | 3.04E−04 | 1.88E−04 |
| Oocyte meiosis | 0.022 | Hepatitis B | 0.003 | Autoimmune thyroid disease | 0.001 | 2.23E−04 |
| Mismatch repair | 0.029 | Cytosolic DNA-sensing pathway | 0.004 | C-type lectin receptor signaling pathway | 0.001 | 0.01 |
| NF-kappa B signaling pathway | 0.031 | Necroptosis | 0.004 | Tuberculosis | 0.002 | 2.01E−04 |
| ECM-receptor interaction | 0.041 | Hepatitis C | 0.005 | Allograft rejection | 0.002 | 2.23E−04 |
| p53 signaling pathway | 0.048 | Bile secretion | 0.007 | Antigen processing and presentation | 0.004 | 1.31E−05 |
| Small cell lung cancer | 0.049 | Hypertrophic cardiomyopathy | 0.007 | Type I diabetes mellitus | 0.004 | 2.23E−04 |
| | | Dilated cardiomyopathy | 0.008 | Graft-*versus*-host disease | 0.004 | 2.23E−04 |
| | | Mineral absorption | 0.008 | Chemokine signaling pathway | 0.004 | 8.82E−04 |
| | | cGMP-PKG signaling pathway | 0.009 | Intestinal immune network for IgA production | 0.006 | 0.007 |
| | | Chagas disease | 0.009 | Complement and coagulation cascades | 0.008 | 0.026 |
| | | Legionellosis | 0.016 | Influenza A | 0.01 | 1.83E−05 |
| | | Herpes simplex virus 1 infection | 0.017 | Viral myocarditis | 0.01 | 8.40E−05 |
| | | Human papillomavirus infection | 0.017 | Osteoclast differentiation | 0.01 | 0.001 |
| | | PI3K-Akt signaling pathway | 0.023 | NOD-like receptor signaling pathway | 0.015 | 2.23E−04 |
| | | AGE-RAGE signaling pathway in diabetic complications | 0.027 | Toll-like receptor signaling pathway | 0.021 | 1.32E−04 |

| Macrophage-specific | | Osteoclast-specific | | Differentiation-independent | | |
|---|---|---|---|---|---|---|
| Pathway | *p*-value | Pathway | *p*-value | Pathway | *p*-value (macrophage) | *p*-value (osteoclast) |
| | | GABAergic synapse | 0.027 | Transcriptional misregulation in cancer | 0.021 | 0.012 |
| | | Oxytocin signaling pathway | 0.027 | TNF signaling pathway | 0.021 | 0.025 |
| | | Arachidonic acid metabolism | 0.03 | Toxoplasmosis | 0.022 | 0.014 |
| | | Salmonella infection | 0.032 | Neuroactive ligand-receptor interaction | 0.028 | 1.88E−04 |
| | | Human immunodeficiency virus 1 infection | 0.038 | Human T-cell leukemia virus 1 infection | 0.028 | 0.002 |
| | | Rap1 signaling pathway | 0.038 | Epstein-Barr virus infection | 0.031 | 4.31E−04 |
| | | Circadian entrainment | 0.04 | Phagosome | 0.041 | 6.41E−04 |
| | | Protein digestion and absorption | 0.04 | Th17 cell differentiation | 0.041 | 0.019 |
| | | Salivary secretion | 0.04 | | | |
| | | Adrenergic signaling in cardiomyocytes | 0.042 | | | |

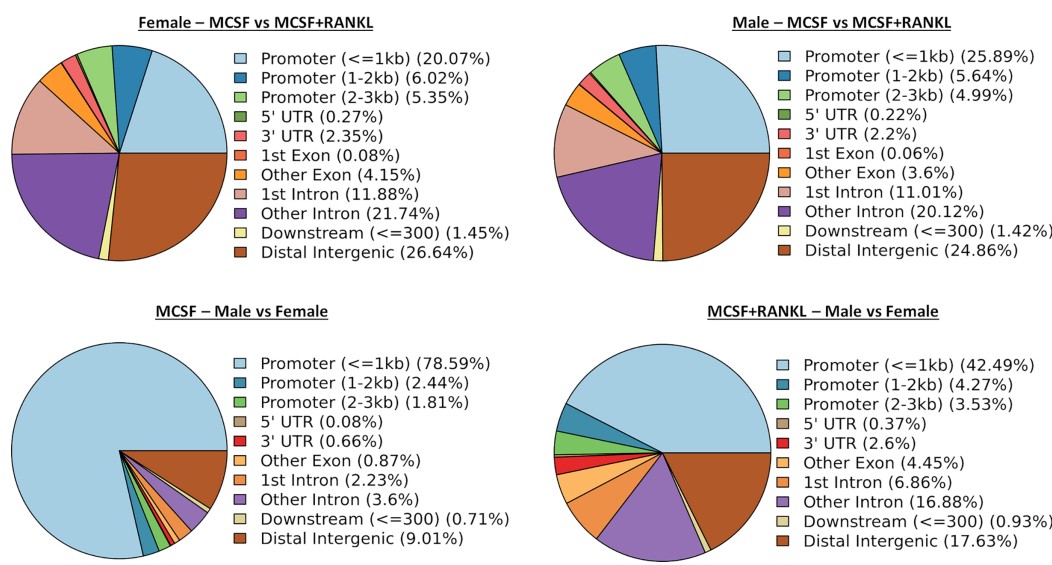

**Figure 6 Differentially accessible chromosome locations in female and male macrophages cultured with MCSF alone or MCSF and RANKL.** Most differential accessible regions were located within promoter regions.

to evaluate pathway-level changes (*St. Laurent et al., 2013*). The top two sex-independent pathways altered by osteoclast differentiation, ranked by FDR-corrected *p*-value, were metabolic pathways and axon guidance. With respect to metabolic pathways, alterations were diverse with most changes falling within pathways of fatty acid metabolism, citric acid

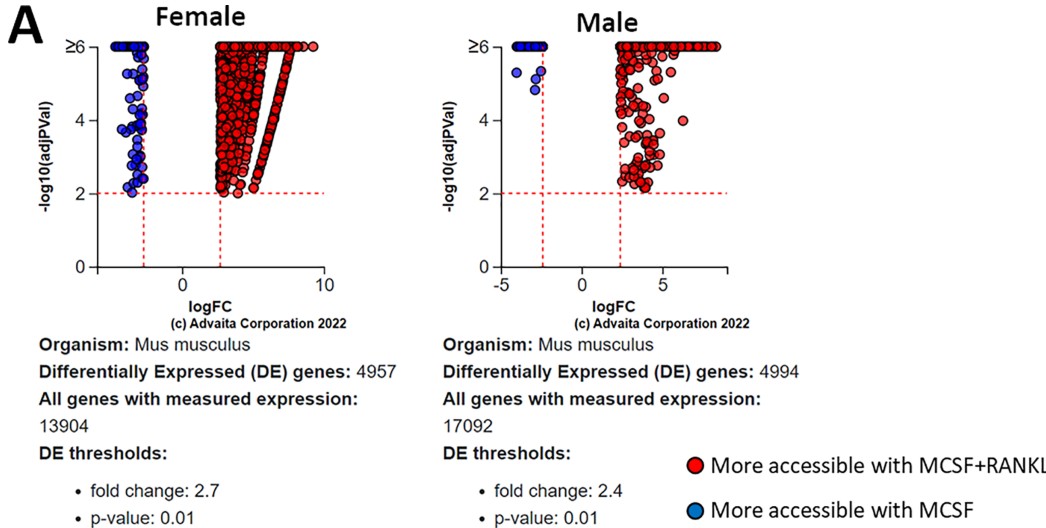

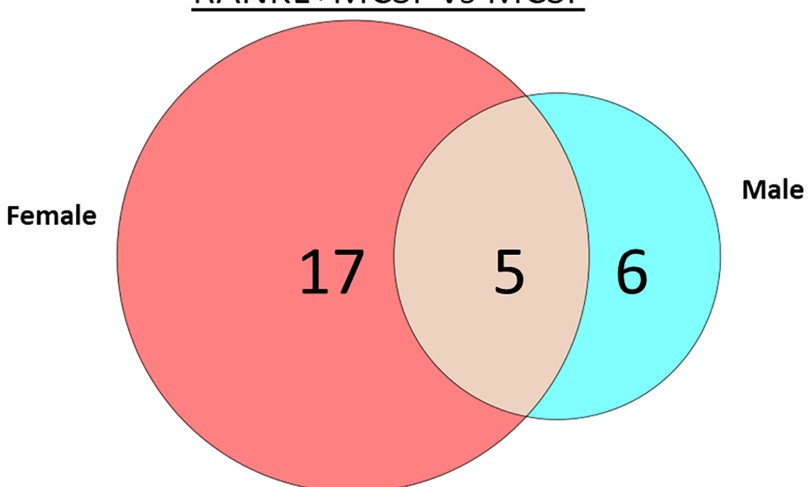

**Figure 7 Gene and pathway analysis of osteoclastogenesis ATAC-Seq data.** (A) Volcano plots depicting genes meeting fold change and *p*-value thresholds in female and male cells. Differentials with −log10 transformed adjusted *p*-values of six and greater are plotted at the same level. Full volcano plots are included in Fig. S1. (B) Female-specific, male-specific, and common pathways altered during osteoclastogenesis. Pathways are listed in Table 3.

cycle, and oxidative phosphorylation. Almost without exception, when gene expression was altered in these pathways, it was increased. This concurs with prior findings of enhanced ATP production and mitochondrial activity in mature osteoclasts (*Li et al., 2020*; *Da, Tao & Zhu, 2021*). Some members of the axon guidance pathway have already been implicated in bone homeostasis such as the Eph/Ephrin and Slit/Robo axes (*Matsuo & Otaki, 2012*; *Kim et al., 2018*, p. 3; *Jiang, Sun & Huang, 2022*). In addition to identifying increased EphA, EphrinA, and EphrinB gene expression and decreased Slit1 expression in

**Table 3 Differential chromatin accessible pathways during osteoclastogenesis.**

| Female-specific | | Male-specific | | Sex-independent | | |
|---|---|---|---|---|---|---|
| Pathway | *p*-value | Pathway | *p*-value | Pathway | *p*-value (Female) | *p*-value (Male) |
| Small cell lung cancer | 0.006 | Protein digestion and absorption | 0.008 | Hippo signaling pathway | 7.52E−04 | 0.004 |
| Endocrine resistance | 0.007 | cGMP-PKG signaling pathway | 0.012 | Axon guidance | 8.12E−04 | 0.005 |
| Prion disease | 0.009 | Pancreatic secretion | 0.018 | Focal adhesion | 0.001 | 0.007 |
| Alpha-Linolenic acid metabolism | 0.012 | Arrhythmogenic right ventricular cardiomyopathy | 0.019 | ECM-receptor interaction | 0.002 | 0.004 |
| Growth hormone synthesis, secretion and action | 0.015 | Ubiquinone and other terpenoid-quinone biosynthesis | 0.031 | Insulin secretion | 0.035 | 0.04 |
| Platinum drug resistance | 0.018 | Aldosterone synthesis and secretion | 0.043 | | | |
| HIF-1 signaling pathway | 0.025 | | | | | |
| Relaxin signaling pathway | 0.025 | | | | | |
| Rap1 signaling pathway | 0.029 | | | | | |
| Fluid shear stress and atherosclerosis | 0.041 | | | | | |
| Glycosaminoglycan biosynthesis—heparan sulfate/heparin | 0.041 | | | | | |
| Linoleic acid metabolism | 0.041 | | | | | |
| Central carbon metabolism in cancer | 0.042 | | | | | |
| Estrogen signaling pathway | 0.042 | | | | | |
| PI3K-Akt signaling pathway | 0.044 | | | | | |
| Proteoglycans in cancer | 0.045 | | | | | |
| Necroptosis | 0.048 | | | | | |

osteoclasts, we found altered expression of other axon guidance membrane receptors including Frizzled3 (Fzd3) and biregional cell adhesion molecule-related/down-regulated by oncogenes binding protein (Boc), which have not been directly interrogated for roles in osteoclast function.

Alterations in pathways associated with immune function was a common finding when comparing male and female-derived cells, which is in agreement with prior published findings (*Gal-Oz et al., 2019*; *Mun et al., 2021*). Interestingly, for multiple pathways linked to immune responses to diverse pathogens, expression of pattern recognition receptors is higher in male macrophages, but, after differentiation, female osteoclasts demonstrate higher expression. In the Toll-like receptor (TLR) pathway, for example, expression of all TLRs except TLR5 (for which there was no significant difference) was higher in female osteoclasts. By contrast, expression of TLR1, TLR2, and TLR9 was higher in male macrophages. These findings suggest that male naïve macrophages are more sensitive to TLR ligands and agree with prior findings of sexually dimorphic responses to inflammatory signals and induction of TLR expression in conjunction with pro-osteoclastogenic pathways (*Marriott, Bost & Huet-Hudson, 2006*; *Li et al., 2009*; *Barcena et al., 2021*; *Chen, Lainez & Coss, 2021*; *Song et al., 2022*). The potency of some

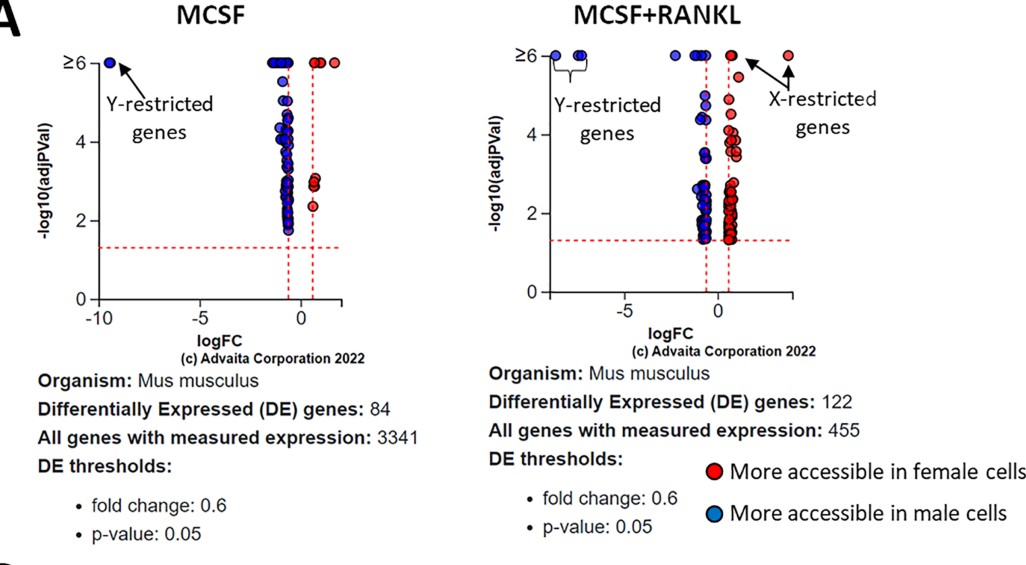

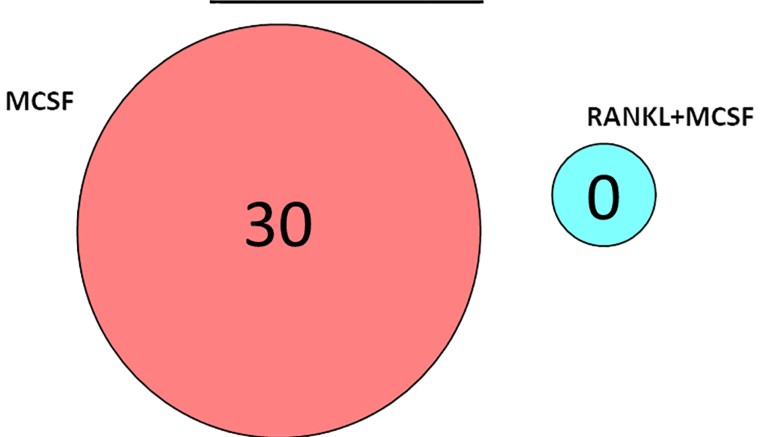

**Figure 8 Gene and pathway analysis of sexually divergent ATAC-Seq data.** (A) Volcano plots depicting genes meeting fold change and *p*-value thresholds in MCSF and MCSF+RANKL treated cells. Differentials with −log10 transformed adjusted *p*-values of six and greater are plotted at the same level. Full volcano plots are included in Fig. S1. (B) MCSF-specific, MCSF+RANKL-specific, and common pathways altered during osteoclastogenesis. Pathways are listed in Table 4.

TLR ligands, such as bacterial lipopolysaccharide (LPS), to stimulate osteoclastogenesis of osteoclast precursors pre-committed with RANKL is well established (*Liu et al., 2009*). It has also been shown that there is a sexually divergent response to LPS by differentiating osteoclasts with female cells demonstrating more robust differentiation (*Mun et al., 2021*).

In addition to pathways with previously established roles in osteoclast function, our analyses identified additional pathways that have not been studied directly in osteoclasts.

**Table 4 Differential chromatin accessible pathways between sexes.**

| MCSF-specific | | MCSF+RANKL specific | Treatment-independent |
|---|---|---|---|
| Pathway | p-value | Pathway | Pathway |
| Influenza A | 2.76E−05 | NONE | NONE |
| Antigen processing and presentation | 4.67E−05 | | |
| Viral myocarditis | 5.94E−05 | | |
| Th17 cell differentiation | 6.01E−05 | | |
| Asthma | 7.99E−05 | | |
| Leishmaniasis | 1.72E−04 | | |
| Staphylococcus aureus infection | 2.97E−04 | | |
| Allograft rejection | 3.92E−04 | | |
| Autoimmune thyroid disease | 3.92E−04 | | |
| Graft-versus-host disease | 3.92E−04 | | |
| Type I diabetes mellitus | 3.92E−04 | | |
| Th1 and Th2 cell differentiation | 4.77E−04 | | |
| Inflammatory bowel disease | 6.85E−04 | | |
| Rheumatoid arthritis | 7.02E−04 | | |
| Intestinal immune network for IgA production | 8.25E−04 | | |
| Toxoplasmosis | 0.002 | | |
| Tuberculosis | 0.002 | | |
| Cell adhesion molecules | 0.003 | | |
| Hematopoietic cell lineage | 0.003 | | |
| Systemic lupus erythematosus | 0.003 | | |
| TGF-beta signaling pathway | 0.004 | | |
| Phagosome | 0.005 | | |
| Transcriptional misregulation in cancer | 0.007 | | |
| Human T-cell leukemia virus 1 infection | 0.011 | | |
| IL-17 signaling pathway | 0.014 | | |
| Rap1 signaling pathway | 0.019 | | |
| Herpes simplex virus 1 infection | 0.031 | | |
| Cholesterol metabolism | 0.033 | | |
| Epstein-Barr virus infection | 0.037 | | |
| ECM-receptor interaction | 0.046 | | |

For example, the Synaptic vesicle cycle (KEGG: 04721) was identified as a sex-independent up-regulated pathway in osteoclastogenesis. While this pathway has also appeared in analyses of differentially expressed genes in post-menopausal osteoporosis, roles of specific differential genes, such as Cplx2, Cacna1b, and Dnm1, have not been investigated (*Zhu et al., 2018*). This pathway and others not currently associated with macrophage or osteoclast function might represent the richest areas for discovery of novel gene functions,

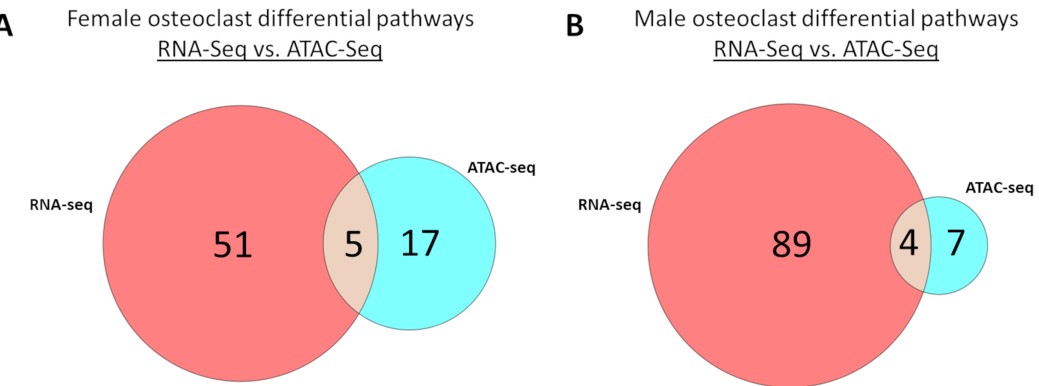

**Figure 9** Correlated pathways identified *via* RNA-Seq and ATAC-Seq. (A) RNA-Seq-specific, ATAC-specific, and common pathways in female cells. Pathways are listed in Table 5. (B) RNA-Seq-specific, ATAC-specific, and common pathways in male cells. Pathways are listed in Table 6.

and we intend to investigate candidate genes of these pathways for deeper insights into osteoclast regulation.

As did the RNA-seq analyses, the ATAC-Seq studies identified expected sexually divergent chromosome accessibility with Y chromosome-specific loci overrepresented in male samples and X chromosome-specific loci overrepresented in female samples. The degree to which RANK signaling loosens chromatin is remarkable, with more accessible genes accounting for greater than 94% of the differences in both sexes. Given the balance of genes that are up- and down-regulated by RANK signaling, a similar pattern of chromatin accessibility might be expected. On the contrary, RANK signaling results in a more transcriptionally accessible epigenetic landscape. This might explain the ability of diverse inflammatory signals to complete osteoclastogenesis in RANKL-committed precursors—RANK signaling might render previously inaccessible pro-osteoclast genes available to transcription factors downstream from inflammatory receptors. By comparison, sex appears to have little influence in chromatin accessibility, suggesting that alterations in gene expression between male and female osteoclast precursors are driven primarily by transcriptional, rather than epigenetic, mechanisms.

Transcription appears to be a greater driver of osteoclastogenesis than epigenetic changes. In both female and male osteoclast precursors, pathway analysis revealed a greater number of significantly altered pathways following RNA-seq than ATAC-seq with minimal overlap between the two. Axon guidance was the only pathway with concordance between RNA-seq and ATAC-seq in both sexes, and while RNA-seq revealed a mixture of up- and down-regulated genes, ATAC-seq demonstrated an almost uniform increase in accessibility. While increased transcription in RANKL-treated cells may depend on increased chromatin accessibility, our data suggest that decreased transcription is not reliant on decreased chromatin accessibility.

**Table 5 Differential transcriptional and chromatin accessible pathways during osteoclastogenesis in female cells.**

| RNA-seq only | | ATAC-seq only | | Common | | |
|---|---|---|---|---|---|---|
| Pathway | p-value | Pathway | p-value | Pathway | p-value (RNA-seq) | p-value (ATAC-Seq) |
| Non-alcoholic fatty liver disease | 2.58E−07 | Hippo signaling pathway | 7.52E−04 | Prion disease | 1.13E−05 | 0.009 |
| Amyotrophic lateral sclerosis | 9.07E−07 | Focal adhesion | 0.001 | Axon guidance | 1.96E−04 | 8.12E−04 |
| Huntington disease | 9.32E−07 | Small cell lung cancer | 0.006 | Estrogen signaling pathway | 0.013 | 0.042 |
| Thermogenesis | 2.79E−06 | Endocrine resistance | 0.007 | Rap1 signaling pathway | 0.026 | 0.029 |
| Retrograde endocannabinoid signaling | 3.10E−06 | Alpha-Linolenic acid metabolism | 0.012 | ECM-receptor interaction | 0.028 | 0.002 |
| Pathways of neurodegeneration-multiple diseases | 6.85E−06 | Growth hormone synthesis, secretion and action | 0.015 | | | |
| Parkinson disease | 6.86E−06 | Platinum drug resistance | 0.018 | | | |
| Cardiac muscle contraction | 7.75E−06 | HIF-1 signaling pathway | 0.025 | | | |
| Citrate cycle (TCA cycle) | 7.75E−06 | Relaxin signaling pathway | 0.025 | | | |
| Collecting duct acid secretion | 7.75E−06 | Insulin secretion | 0.035 | | | |
| Metabolic pathways | 7.75E−06 | Fluid shear stress and atherosclerosis | 0.041 | | | |
| Oxidative phosphorylation | 7.75E−06 | Glycosaminoglycan biosynthesis—heparan sulfate/heparin | 0.041 | | | |
| Alzheimer disease | 1.18E−05 | Linoleic acid metabolism | 0.041 | | | |
| Synaptic vesicle cycle | 3.37E−05 | Central carbon metabolism in cancer | 0.042 | | | |
| Phagosome | 6.04E−05 | PI3K-Akt signaling pathway | 0.044 | | | |
| Rheumatoid arthritis | 9.49E−05 | Proteoglycans in cancer | 0.045 | | | |
| Calcium signaling pathway | 1.39E−04 | Necroptosis | 0.048 | | | |
| Neuroactive ligand-receptor interaction | 1.39E−04 | | | | | |
| Cell adhesion molecules | 3.59E−04 | | | | | |
| Taste transduction | 0.001 | | | | | |
| Systemic lupus erythematosus | 0.002 | | | | | |
| Arrhythmogenic right ventricular cardiomyopathy | 0.004 | | | | | |
| cAMP signaling pathway | 0.004 | | | | | |
| Pyruvate metabolism | 0.004 | | | | | |
| Regulation of actin cytoskeleton | 0.005 | | | | | |
| Carbon metabolism | 0.006 | | | | | |
| *Staphylococcus aureus* infection | 0.006 | | | | | |
| Complement and coagulation cascades | 0.008 | | | | | |
| Dilated cardiomyopathy | 0.008 | | | | | |
| Viral myocarditis | 0.008 | | | | | |
| Hypertrophic cardiomyopathy | 0.013 | | | | | |
| Hematopoietic cell lineage | 0.017 | | | | | |

| RNA-seq only | | ATAC-seq only | | Common | | |
| --- | --- | --- | --- | --- | --- | --- |
| Pathway | *p*-value | Pathway | *p*-value | Pathway | *p*-value (RNA-seq) | *p*-value (ATAC-Seq) |
| Pancreatic secretion | 0.017 | | | | | |
| Transcriptional misregulation in cancer | 0.017 | | | | | |
| Cytokine-cytokine receptor interaction | 0.021 | | | | | |
| Pathways in cancer | 0.024 | | | | | |
| Chemokine signaling pathway | 0.027 | | | | | |
| ECM-receptor interaction | 0.028 | | | | | |
| Arachidonic acid metabolism | 0.03 | | | | | |
| Gap junction | 0.031 | | | | | |
| Herpes simplex virus 1 infection | 0.031 | | | | | |
| GABAergic synapse | 0.033 | | | | | |
| Glycosaminoglycan degradation | 0.033 | | | | | |
| Glycosphingolipid biosynthesis - lacto and neolacto series | 0.033 | | | | | |
| Cellular senescence | 0.037 | | | | | |
| Glutathione metabolism | 0.038 | | | | | |
| p53 signaling pathway | 0.038 | | | | | |
| Gastric acid secretion | 0.039 | | | | | |
| Glycolysis/Gluconeogenesis | 0.047 | | | | | |
| Phospholipase D signaling pathway | 0.048 | | | | | |
| Progesterone-mediated oocyte maturation | 0.048 | | | | | |
| Cholinergic synapse | 0.05 | | | | | |

**Table 6 Differential transcriptional and chromatin accessible pathways during osteoclastogenesis in male cells.**

| RNA-seq only | | ATAC-seq only | | Common | | |
| --- | --- | --- | --- | --- | --- | --- |
| Pathway | *p*-value | Pathway | *p*-value | Pathway | *p*-value (RNA-Seq) | *p*-value (ATAC-Seq) |
| Cytokine-cytokine receptor interaction | 1.12E−08 | ECM-receptor interaction | 0.004 | Arrhythmogenic right ventricular cardiomyopathy | 0.031 | 0.019 |
| Viral protein interaction with cytokine and cytokine receptor | 1.12E−08 | Hippo signaling pathway | 0.004 | Axon guidance | 5.41E−04 | 0.005 |
| Antigen processing and presentation | 1.18E−07 | Focal adhesion | 0.007 | cGMP-PKG signaling pathway | 0.009 | 0.012 |
| *Staphylococcus aureus* infection | 2.12E−07 | Protein digestion and absorption | 0.008 | Pancreatic secretion | 0.013 | 0.018 |

| RNA-seq only | | ATAC-seq only | | Common | | |
|---|---|---|---|---|---|---|
| Pathway | *p*-value | Pathway | *p*-value | Pathway | *p*-value (RNA-Seq) | *p*-value (ATAC-Seq) |
| Systemic lupus erythematosus | 2.87E−07 | Ubiquinone and other terpenoid-quinone biosynthesis | 0.031 | | | |
| Influenza A | 3.74E−07 | Insulin secretion | 0.04 | | | |
| Neuroactive ligand-receptor interaction | 1.45E−06 | Aldosterone synthesis and secretion | 0.043 | | | |
| Type I diabetes mellitus | 1.64E−06 | | | | | |
| Toll-like receptor signaling pathway | 1.70E−06 | | | | | |
| Graft-*versus*-host disease | 1.74E−06 | | | | | |
| Allograft rejection | 1.78E−06 | | | | | |
| Intestinal immune network for IgA production | 1.88E−06 | | | | | |
| Non-alcoholic fatty liver disease | 2.19E−06 | | | | | |
| Autoimmune thyroid disease | 2.76E−06 | | | | | |
| Prion disease | 3.13E−06 | | | | | |
| Inflammatory bowel disease | 3.47E−06 | | | | | |
| Chemokine signaling pathway | 4.75E−06 | | | | | |
| Rheumatoid arthritis | 4.87E−06 | | | | | |
| Viral myocarditis | 7.22E−06 | | | | | |
| Cardiac muscle contraction | 7.75E−06 | | | | | |
| Cell adhesion molecules | 7.75E−06 | | | | | |
| Citrate cycle (TCA cycle) | 7.75E−06 | | | | | |
| Collecting duct acid secretion | 7.75E−06 | | | | | |
| Hematopoietic cell lineage | 7.75E−06 | | | | | |
| Metabolic pathways | 7.75E−06 | | | | | |
| Oxidative phosphorylation | 7.75E−06 | | | | | |
| Phagosome | 7.75E−06 | | | | | |
| Malaria | 8.15E−06 | | | | | |
| Tuberculosis | 1.23E−05 | | | | | |
| Asthma | 1.41E−05 | | | | | |
| Calcium signaling pathway | 2.95E−05 | | | | | |
| Thermogenesis | 3.67E−05 | | | | | |
| Measles | 3.70E−05 | | | | | |
| Synaptic vesicle cycle | 3.80E−05 | | | | | |
| NOD-like receptor signaling pathway | 4.74E−05 | | | | | |
| Bile secretion | 1.81E−04 | | | | | |
| Pertussis | 2.52E−04 | | | | | |
| Osteoclast differentiation | 2.97E−04 | | | | | |

| RNA-seq only | | ATAC-seq only | | Common | | |
|---|---|---|---|---|---|---|
| Pathway | *p*-value | Pathway | *p*-value | Pathway | *p*-value (RNA-Seq) | *p*-value (ATAC-Seq) |
| Hepatitis B | 5.19E−04 | | | | | |
| Human cytomegalovirus infection | 6.35E−04 | | | | | |
| Parkinson disease | 7.15E−04 | | | | | |
| Retrograde endocannabinoid signaling | 8.66E−04 | | | | | |
| Legionellosis | 8.87E−04 | | | | | |
| Alzheimer disease | 0.001 | | | | | |
| Arachidonic acid metabolism | 0.001 | | | | | |
| cAMP signaling pathway | 0.001 | | | | | |
| Huntington disease | 0.001 | | | | | |
| Pathways in cancer | 0.001 | | | | | |
| Toxoplasmosis | 0.001 | | | | | |
| C-type lectin receptor signaling pathway | 0.002 | | | | | |
| Lysosome | 0.002 | | | | | |
| MAPK signaling pathway | 0.002 | | | | | |
| Th1 and Th2 cell differentiation | 0.002 | | | | | |
| Th17 cell differentiation | 0.002 | | | | | |
| Necroptosis | 0.003 | | | | | |
| Pyruvate metabolism | 0.003 | | | | | |
| ABC transporters | 0.004 | | | | | |
| Human papillomavirus infection | 0.004 | | | | | |
| Epstein-Barr virus infection | 0.007 | | | | | |
| Herpes simplex virus 1 infection | 0.008 | | | | | |
| Rap1 signaling pathway | 0.008 | | | | | |
| NF-kappa B signaling pathway | 0.009 | | | | | |
| Pathways of neurodegeneration—multiple diseases | 0.009 | | | | | |
| TNF signaling pathway | 0.009 | | | | | |
| Complement and coagulation cascades | 0.011 | | | | | |
| Leishmaniasis | 0.016 | | | | | |
| Salmonella infection | 0.016 | | | | | |
| Estrogen signaling pathway | 0.017 | | | | | |
| Mineral absorption | 0.017 | | | | | |
| Regulation of actin cytoskeleton | 0.021 | | | | | |
| Chagas disease | 0.022 | | | | | |
| JAK-STAT signaling pathway | 0.023 | | | | | |

| RNA-seq only | | ATAC-seq only | | Common | | |
|---|---|---|---|---|---|---|
| Pathway | *p*-value | Pathway | *p*-value | Pathway | *p*-value (RNA-Seq) | *p*-value (ATAC-Seq) |
| Human immunodeficiency virus 1 infection | 0.026 | | | | | |
| Salivary secretion | 0.026 | | | | | |
| Carbon metabolism | 0.027 | | | | | |
| Dilated cardiomyopathy | 0.028 | | | | | |
| Gastric acid secretion | 0.029 | | | | | |
| Taste transduction | 0.029 | | | | | |
| Proximal tubule bicarbonate reclamation | 0.03 | | | | | |
| Glycosaminoglycan degradation | 0.031 | | | | | |
| Transcriptional misregulation in cancer | 0.031 | | | | | |
| Sphingolipid metabolism | 0.032 | | | | | |
| Primary bile acid biosynthesis | 0.04 | | | | | |
| Morphine addiction | 0.043 | | | | | |
| Inflammatory mediator regulation of TRP channels | 0.044 | | | | | |
| African trypanosomiasis | 0.045 | | | | | |
| Ras signaling pathway | 0.047 | | | | | |
| Phospholipase D signaling pathway | 0.048 | | | | | |
| Proteoglycans in cancer | 0.049 | | | | | |

## CONCLUSIONS

In this study we hypothesized that gene expression patterns and chromatin accessibility in osteoclast precursors are regulated by both RANK signaling and sex-specific factors. Using RNA-seq, ATAC-seq, and pathway analyses, we found that RANKL produces different transcriptional patterns in male- and female-derived cells, and these patterns suggest that male and female osteoclast precursors may respond differently to inflammatory signals. Furthermore, we found that RANK signaling results in a more permissive epigenetic landscape, with most loci becoming more accessible following treatment with RANKL. Multiple studies have demonstrated that inflammatory signals including LPS, poly I:C, and TNF can support the differentiation of committed osteoclast precursors while those same factors prevent osteoclastogenesis of naïve precursors. Our data suggest that male and female osteoclast precursors may have different sensitivities to these and other factors. Future studies should investigate these potential differences, which may further explain sexually divergent risk of bone loss.

### Funding

This work was supported by a Faculty Grant in Research and Creative Works (FGRCW) from Eastern Washington University. The funders had no role in study design, data collection and analysis, decision to publish, or preparation of the manuscript.

### Grant Disclosures

The following grant information was disclosed by the authors:
Eastern Washington University.

### Competing Interests

The authors declare that they have no competing interests.

### Author Contributions

- Abigail L. Keever conceived and designed the experiments, performed the experiments, analyzed the data, prepared figures and/or tables, authored or reviewed drafts of the article, and approved the final draft.
- Kathryn M. Collins performed the experiments, authored or reviewed drafts of the article, and approved the final draft.
- Rachel A. Clark performed the experiments, authored or reviewed drafts of the article, and approved the final draft.
- Amber L. Framstad performed the experiments, authored or reviewed drafts of the article, and approved the final draft.
- Jason W. Ashley conceived and designed the experiments, performed the experiments, analyzed the data, prepared figures and/or tables, authored or reviewed drafts of the article, and approved the final draft.

### Data Availability

  The RNA sequencing data is available at NCBI GEO: GSE216929.

### Supplemental Information

Supplemental information for this article can be found online at http://dx.doi.org/10.7717/peerj.14814#supplemental-information.

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
