# Peer review of "RANK signaling in osteoclast precursors results in a more permissive epigenetic landscape and sexually divergent patterns of gene expression"

_PeerJ, doi:10.7717/peerj.14814_

## Round 0.1 · original submission · Major Revisions

The reviewers have several queries in relation to the experimental design and specific changes needed with regard to data presentation.

Reviewer 1 ·

Basic reporting

This work explores the global differential expression and chromatic accessibility in macrophages and osteoclasts of male and female C67BL/6 mice. By using RNA-seq and ATAC-seq analyses, the authors report that there are significant differences in both female and male osteoclasts as compared to macrophages and, furthermore, bioinformatic analysis of the related functions revealed that male macrophages and female osteoclasts show greater sensitivity in immunological pathways. Finally, ATAC-seq demonstrated that RANKL addition can induce chromatin accessibility independent of the gender.

Overall, the manuscript is well written. However, there are some issues with the presentation of the results and the authors need to address the following points to improve the quality of the manuscript.

Experimental design

The study design is acceptable.

• The authors state that mice were 2-6 months old. However, it is well known that the difference between a 2-months and a 6-months old mouse in remarkable in terms of skeletal maturity and bone mass; A female C57BL/6 mouse has lost more that half of the bone mass at 6-months as compared to a 2-months mouse and this trend is also followed in male mice at a lower degree. This needs clarification. In addition, it is not clear many mice were used.
• For both RNA- and ATAC-seq, essential details for the analysis are missing. For example, is there any counts threshold, did the authors use R etc? The exact steps for de-multiplexing, alignment, transcript identification, and differential expression analysis for RNA-seq must be provided and the same stands for the ATAc-seq.

Validity of the findings

The authors need to improve the presentation of the results.

• The authors claim that RANKL and MCSF addition resulted in large, multinucleated osteoclasts which are shown in Fig1. This has to be verified by TRAP staining.
• In the results and the Fig3 legends the authors wrote that “MMP9, Ctsk, and Acp5 demonstrate the largest differences in expression”. Is this based on the padj? Or FC? Or counts? Looking at the corresponding spreadsheet in the supplemental material, this is confusing. For example, when male osteoclasts and macrophages are compared, there are genes with higher DE, based on this spreadsheet.
• What data did the authors use to produce the Volcano plots? In Fig4A, the highlighted gene, Slco2b1 has a padj of 2.81E-271 so the -log10 is approx. 270. All the plots need to be carefully checked.
• In Fig5, the FC threshold is 0.5-0.6. How safe are the conclusions from this part?
• The p values of all significant pathways must be included. In addition, many of these pathways are irrelevant to the nature of the cell types. Please comment.

Additional comments

• A deeper discussion with support from the current literature would add to the manuscript.

Reviewer 2 ·

Basic reporting

pass

Experimental design

pass

Validity of the findings

pass

Additional comments

In this manuscript the authors examine differences in gene expression and chromatin accessibility between cultured murine macrophages and osteoclast and the sex of the mouse from which the cells originated. Studies were performed on differentiating murine bone marrow macrophage cultures after 3 days of culture with either M-CSF alone or M-CSF + RANKL. Studies relied on bulk RNA-Seq and ATAC-Seq from 3 day BMM cultures that were treated with either M-CSF alone to produce macrophages or M-CSF + RANKL to commit cells to the osteoclast lineage. The authors find that there are major differences in gene expression between macrophages and osteoclasts and whether the cells originated from male of female mice. In general, male macrophages had higher expression of toll-like receptors (TLRs) while the opposite was true for osteoclasts. There were not major differences in chromatin accessibility between males and female osteoclasts, However, induction of the osteoclast lineage by treatment of BMMs with RANKL induced a major enhancement. As previously demonstrated by Mun et al, differences in osteoclast gene expression between females and males were mainly due to increased expression of inflammatory genes in female osteoclasts.

Overall, this is a solid work. The analysis of the gene expression and chromatin accessibility is well done and extensive. Conclusions seem well founded; given the results of the studies and the ATAC seq data is novel. My critique only highlights minor points:

1) The authors state in line 282 that Liu, et al, 2009, demonstrated differences in osteoclastogenesis in response to LPS in male and female differentiating BMMs. However, my quick perusal of Liu, et al. could find no mention that this paper examined sexual dimorphic osteoclastogenic responses to LPS. What did I miss?

2) The authors conclusion that differences in gene expression in differentiating osteoclast cultures between female-origin and male-origin cells were due to enhanced inflammatory gene expression in female-origin cells is similar to that of Mun et al., 2021 (see the title of that manuscript in the references) . The authors should state this fact in the discussion.

---

## Round 0.2 · accepted · Accept

The reviewers have evaluated the corrected manuscript and are satisfied with the improvements made. On that basis, the reviewers have deemed your manuscript suitable for publication.

Reviewer 1 ·

Basic reporting

The authors addressed me comments.

Experimental design

Methods were improved and clarifications provided by a sufficient manner.

Validity of the findings

The presentation and accuracy of the results have been improved.

Reviewer 2 ·

Basic reporting

OK with me.

Experimental design

OK with me.

Validity of the findings

OK with me.

Additional comments

none